# Low-rank matrix reconstruction and clustering via approximate message passing

**Ryosuke Matsushita**
NTT DATA Mathematical Systems Inc.
1F Shinanomachi Rengakan, 35,
Shinanomachi, Shinjuku-ku, Tokyo,
160-0016, Japan
matsur8@gmail.com

**Toshiyuki Tanaka**
Department of Systems Science,
Graduate School of Informatics, Kyoto University
Yoshida Hon-machi, Sakyo-ku, Kyoto-shi,
606-8501 Japan
tt@i.kyoto-u.ac.jp

## Abstract

We study the problem of reconstructing low-rank matrices from their noisy observations. We formulate the problem in the Bayesian framework, which allows us to exploit structural properties of matrices in addition to low-rankedness, such as sparsity. We propose an efficient approximate message passing algorithm, derived from the belief propagation algorithm, to perform the Bayesian inference for matrix reconstruction. We have also successfully applied the proposed algorithm to a clustering problem, by reformulating it as a low-rank matrix reconstruction problem with an additional structural property. Numerical experiments show that the proposed algorithm outperforms Lloyd's K-means algorithm.

## 1 Introduction

Low-rankedness of matrices has frequently been exploited when one reconstructs a matrix from its noisy observations. In such problems, there are often demands to incorporate additional structural properties of matrices in addition to the low-rankedness. In this paper, we consider the case where a matrix $A_0 \in \mathbb{R}^{m \times N}$ to be reconstructed is factored as $A_0 = U_0 V_0^\top$, $U_0 \in \mathbb{R}^{m \times r}$, $V_0 \in \mathbb{R}^{N \times r}$ ($r \ll m$, $N$), and where one knows structural properties of the factors $U_0$ and $V_0$ a priori. Sparseness and non-negativity of the factors are popular examples of such structural properties [1, 2].

Since the properties of the factors to be exploited vary according to the problem, it is desirable that a reconstruction method has enough flexibility to incorporate a wide variety of properties. The Bayesian approach achieves such flexibility by allowing us to select prior distributions of $U_0$ and $V_0$ reflecting a priori knowledge on the structural properties. The Bayesian approach, however, often involves computationally expensive processes such as high-dimensional integrations, thereby requiring approximate inference methods in practical implementations. Monte Carlo sampling methods and variational Bayes methods have been proposed for low-rank matrix reconstruction to meet this requirement [3–5].

We present in this paper an approximate message passing (AMP) based algorithm for Bayesian low-rank matrix reconstruction. Developed in the context of compressed sensing, the AMP algorithm reconstructs sparse vectors from their linear measurements with low computational cost, and achieves a certain theoretical limit [6]. AMP algorithms can also be used for approximating Bayesian inference with a large class of prior distributions of signal vectors and noise distributions [7]. These successes of AMP algorithms motivate the use of the same idea for low-rank matrix reconstruction. The IterFac algorithm for the rank-one case [8] has been derived as an AMP algorithm. An AMP algorithm for the general-rank case is proposed in [9], which, however, can only treat estimation of posterior means. We extend their algorithm so that one can deal with other estimations such as the maximum a posteriori (MAP) estimation. It is the first contribution of this paper.

As the second contribution, we apply the derived AMP algorithm to K-means type clustering to obtain a novel efficient clustering algorithm. It is based on the observation that our formulation of the low-rank matrix reconstruction problem includes the clustering problem as a special case. Although the idea of applying low-rank matrix reconstruction to clustering is not new [10, 11], our proposed algorithm is, to our knowledge, the first that directly deals with the constraint that each datum should be assigned to exactly one cluster in the framework of low-rank matrix reconstruction. We present results of numerical experiments, which show that the proposed algorithm outperforms Lloyd's K-means algorithm [12] when data are high-dimensional.

Recently, AMP algorithms for dictionary learning and blind calibration [13] and for matrix reconstruction with a generalized observation model [14] were proposed. Although our work has some similarities to these studies, it differs in that we fix the rank $r$ rather than the ratio $r/m$ when taking the limit $m, N \to \infty$ in the derivation of the algorithm. Another difference is that our formulation, explained in the next section, does not assume statistical independence among the components of each row of $U_0$ and $V_0$. A detailed comparison among these algorithms remains to be made.

## 2 Problem setting

### 2.1 Low-rank matrix reconstruction

We consider the following problem setting. A matrix $A_0 \in \mathbb{R}^{m \times N}$ to be estimated is defined by two matrices $U_0 := (\boldsymbol{u}_{0,1}, \ldots, \boldsymbol{u}_{0,m})^\top \in \mathbb{R}^{m \times r}$ and $V_0 := (\boldsymbol{v}_{0,1}, \ldots, \boldsymbol{v}_{0,N})^\top \in \mathbb{R}^{N \times r}$ as $A_0 := U_0 V_0^\top$, where $\boldsymbol{u}_{0,i}, \boldsymbol{v}_{0,j} \in \mathbb{R}^r$. We consider the case where $r \ll m, N$. Observations of $A_0$ are corrupted by additive noise $W \in \mathbb{R}^{m \times N}$, whose components $W_{i,j}$ are i.i.d. Gaussian random variables following $N(0, m\tau)$. Here $\tau > 0$ is a noise variance parameter and $N(a, \sigma^2)$ denotes the Gaussian distribution with mean $a$ and variance $\sigma^2$. The factor $m$ in the noise variance is introduced to allow a proper scaling in the limit where $m$ and $N$ go to infinity in the same order, which is employed in deriving the algorithm. An observed matrix $A \in \mathbb{R}^{m \times N}$ is given by $A := A_0 + W$. Reconstructing $A_0$ and $(U_0, V_0)$ from $A$ is the problem considered in this paper.

We take the Bayesian approach to address this problem, in which one requires prior distributions of variables to be estimated, as well as conditional distributions relating observations with variables to be estimated. These distributions need not be the true ones because in some cases they are not available so that one has to assume them arbitrarily, and in some other cases one expects advantages by assuming them in some specific manner in view of computational efficiencies. In this paper, we suppose that one uses the true conditional distribution

$$p(A|U_0, V_0) = \frac{1}{(2\pi m\tau)^{\frac{mN}{2}}} \exp\Big(-\frac{1}{2m\tau}\|A - U_0 V_0^\top\|_F^2\Big), \tag{1}$$

where $\|\cdot\|_F$ denotes the Frobenius norm. Meanwhile, we suppose that the assumed prior distributions of $U_0$ and $V_0$, denoted by $\hat{p}_U$ and $\hat{p}_V$, respectively, may be different from the true distributions $p_U$ and $p_V$, respectively. We restrict $\hat{p}_U$ and $\hat{p}_V$ to distributions of the form $\hat{p}_U(U_0) = \prod_i \hat{p}_{\mathbf{u}}(\boldsymbol{u}_{0,i})$ and $\hat{p}_V(V_0) = \prod_j \hat{p}_{\mathbf{v}}(\boldsymbol{v}_{0,j})$, respectively, which allows us to construct computationally efficient algorithms. When $U \sim \hat{p}_U(U)$ and $V \sim \hat{p}_V(V)$, the posterior distribution of $(U, V)$ given $A$ is

$$\hat{p}(U, V|A) \propto \exp\Big(-\frac{1}{2m\tau}\|A - UV^\top\|_F^2\Big)\hat{p}_U(U)\hat{p}_V(V). \tag{2}$$

Prior probability density functions (p.d.f.s) $\hat{p}_{\mathbf{u}}$ and $\hat{p}_{\mathbf{v}}$ can be improper, that is, they can integrate to infinity, as long as the posterior p.d.f. (2) is proper. We also consider cases where the assumed rank $\hat{r}$ may be different from the true rank $r$. We thus suppose that estimates $U$ and $V$ are of size $m \times \hat{r}$ and $N \times \hat{r}$, respectively.

We consider two problems appearing in the Bayesian approach. The first problem, which we call the marginalization problem, is to calculate the marginal posterior distributions given $A$,

$$\hat{p}_{i,j}(\boldsymbol{u}_i, \boldsymbol{v}_j|A) := \int \hat{p}(U, V|A) \prod_{k \neq i} \mathrm{d}\boldsymbol{u}_k \prod_{l \neq j} \mathrm{d}\boldsymbol{v}_l. \tag{3}$$

These are used to calculate the posterior mean $E[UV^\top|A]$ and the marginal MAP estimates $\boldsymbol{u}_i^{\mathrm{MMAP}} := \arg\max_{\boldsymbol{u}} \int \hat{p}_{i,j}(\boldsymbol{u}, \boldsymbol{v}|A)\mathrm{d}\boldsymbol{v}$ and $\boldsymbol{v}_j^{\mathrm{MMAP}} := \arg\max_{\boldsymbol{v}} \int \hat{p}_{i,j}(\boldsymbol{u}, \boldsymbol{v}|A)\mathrm{d}\boldsymbol{u}$. Because

calculation of $\hat{p}_{i,j}(\boldsymbol{u}_i, \boldsymbol{v}_j | A)$ typically involves high-dimensional integrations requiring high computational cost, approximation methods are needed.

The second problem, which we call the MAP problem, is to calculate the MAP estimate $\arg\max_{U,V} \hat{p}(U, V | A)$. It is formulated as the following optimization problem:

$$\min_{U,V} C^{\mathrm{MAP}}(U, V), \tag{4}$$

where $C^{\mathrm{MAP}}(U, V)$ is the negative logarithm of (2):

$$C^{\mathrm{MAP}}(U, V) := \frac{1}{2m\tau} \|A - UV^{\top}\|_F^2 - \sum_{i=1}^{m} \log \hat{p}_{\mathbf{u}}(\boldsymbol{u}_i) - \sum_{j=1}^{N} \log \hat{p}_{\mathbf{v}}(\boldsymbol{v}_j). \tag{5}$$

Because $\|A - UV^{\top}\|_F^2$ is a non-convex function of $(U, V)$, it is generally hard to find the global optimal solutions of (4) and therefore approximation methods are needed in this problem as well.

## 2.2 Clustering as low-rank matrix reconstruction

A clustering problem can be formulated as a problem of low-rank matrix reconstruction [11]. Suppose that $\boldsymbol{v}_{0,j} \in \{\boldsymbol{e}_1, \ldots, \boldsymbol{e}_r\}$, $j = 1, \ldots, N$, where $\boldsymbol{e}_l \in \{0, 1\}^r$ is the vector whose $l$th component is 1 and the others are 0. When $V_0$ and $U_0$ are fixed, $\boldsymbol{a}_j$ follows one of the $r$ Gaussian distributions $N(\tilde{\boldsymbol{u}}_{0,l}, m\tau I)$, $l = 1, \ldots, r$, where $\tilde{\boldsymbol{u}}_{0,l}$ is the $l$th column of $U_0$. We regard that each Gaussian distribution defines a cluster, $\tilde{\boldsymbol{u}}_{0,l}$ being the center of cluster $l$ and $\boldsymbol{v}_{0,j}$ representing the cluster assignment of the datum $\boldsymbol{a}_j$. One can then perform clustering on the dataset $\{\boldsymbol{a}_1, \ldots, \boldsymbol{a}_N\}$ by reconstructing $U_0$ and $V_0$ from $A = (\boldsymbol{a}_1, \ldots, \boldsymbol{a}_N)$ under the structural constraint that every row of $V_0$ should belong to $\{\boldsymbol{e}_1, \ldots, \boldsymbol{e}_{\hat{r}}\}$, where $\hat{r}$ is an assumed number of clusters.

Let us consider maximum likelihood estimation $\arg\max_{U,V} p(A | U, V)$, or equivalently, MAP estimation with the (improper) uniform prior distributions $\hat{p}_{\mathbf{u}}(\boldsymbol{u}) = 1$ and $\hat{p}_{\mathbf{v}}(\boldsymbol{v}) = \hat{r}^{-1} \sum_{l=1}^{\hat{r}} \delta(\boldsymbol{v} - \boldsymbol{e}_l)$. The corresponding MAP problem is

$$\min_{U \in \mathbb{R}^{m \times \hat{r}}, V \in \{0,1\}^{N \times \hat{r}}} \|A - UV^{\top}\|_F^2 \qquad \text{subject to} \quad \boldsymbol{v}_j \in \{\boldsymbol{e}_1, \ldots, \boldsymbol{e}_{\hat{r}}\}. \tag{6}$$

When $V$ satisfies the constraints, the objective function $\|A - UV^{\top}\|_F^2 = \sum_{j=1}^{N} \sum_{l=1}^{\hat{r}} \|\boldsymbol{a}_j - \tilde{\boldsymbol{u}}_l\|_2^2 I(\boldsymbol{v}_j = \boldsymbol{e}_l)$ is the sum of squared distances, each of which is between a datum and the center of the cluster that the datum is assigned to. The optimization problem (6), its objective function, and clustering based on it are called in this paper the K-means problem, the K-means loss function, and the K-means clustering, respectively.

One can also use the marginal MAP estimation for clustering. If $U_0$ and $V_0$ follow $\hat{p}_U$ and $\hat{p}_V$, respectively, the marginal MAP estimation is optimal in the sense that it maximizes the expectation of accuracy with respect to $\hat{p}(V_0 | A)$. Here, accuracy is defined as the fraction of correctly assigned data among all data. We call the clustering using approximate marginal MAP estimation *the maximum accuracy clustering*, even when incorrect prior distributions are used.

## 3 Previous work

Existing methods for approximately solving the marginalization problem and the MAP problem are divided into stochastic methods such as Markov-Chain Monte-Carlo methods and deterministic ones. A popular deterministic method is to use the variational Bayesian formalism. The variational Bayes matrix factorization [4, 5] approximates the posterior distribution $p(U, V | A)$ as the product of two functions $p_U^{\mathrm{VB}}(U)$ and $p_V^{\mathrm{VB}}(V)$, which are determined so that the Kullback-Leibler (KL) divergence from $p_U^{\mathrm{VB}}(U) p_V^{\mathrm{VB}}(V)$ to $p(U, V | A)$ is minimized. Global minimization of the KL divergence is difficult except for some special cases [15], so that an iterative method to obtain a local minimum is usually adopted. Applying the variational Bayes matrix factorization to the MAP problem, one obtains the iterated conditional modes (ICM) algorithm, which alternates minimization of $C^{\mathrm{MAP}}(U, V)$ over $U$ for fixed $V$ and minimization over $V$ for fixed $U$.

The representative algorithm to solve the K-means problem approximately is Lloyd's K-means algorithm [12]. Lloyd's K-means algorithm is regarded as the ICM algorithm: It alternates minimization of the K-means loss function over $U$ for fixed $V$ and minimization over $V$ for fixed $U$ iteratively.

**Algorithm 1** (Lloyd's K-means algorithm)**.**

$$n_l^t = \sum_{j=1}^{N} I(\boldsymbol{v}_j^t = \boldsymbol{e}_l), \qquad \tilde{\boldsymbol{u}}_l^t = \frac{1}{n_l^t} \sum_{j=1}^{N} \boldsymbol{a}_j I(\boldsymbol{v}_j^t = \boldsymbol{e}_l), \tag{7a}$$

$$l_j^{t+1} = \arg \min_{l \in \{1,\dots,\hat{r}\}} \|\boldsymbol{a}_j - \tilde{\boldsymbol{u}}_l^t\|_2^2, \qquad \boldsymbol{v}_j^{t+1} = \boldsymbol{e}_{l_j^{t+1}}. \tag{7b}$$

Throughout this paper, we represent an algorithm by a set of equations as in the above. This representation means that the algorithm begins with a set of initial values and repeats the update of the variables using the equations presented until it satisfies some stopping criteria. Lloyd's K-means algorithm begins with a set of initial assignments $V^0 \in \{\boldsymbol{e}_1,\dots,\boldsymbol{e}_{\hat{r}}\}^N$. This algorithm easily gets stuck in local minima and its performance heavily depends on the initial values of the algorithm. Some methods for initialization to obtain a better local minimum are proposed [16].

Maximum accuracy clustering can be solved approximately by using the variational Bayes matrix factorization, since it gives an approximation to the marginal posterior distribution of $\boldsymbol{v}_j$ given $A$.

# 4 Proposed algorithm

## 4.1 Approximate message passing algorithm for low-rank matrix reconstruction

We first discuss the general idea of the AMP algorithm and advantages of the AMP algorithm compared with the variational Bayes matrix factorization. The AMP algorithm is derived by approximating the belief propagation message passing algorithm in a way thought to be asymptotically exact for large-scale problems with appropriate randomness. Fixed points of the belief propagation message passing algorithm correspond to local minima of the KL divergence between a kind of trial function and the posterior distribution [17]. Therefore, the belief propagation message passing algorithm can be regarded as an iterative algorithm based on an approximation of the posterior distribution, which is called the Bethe approximation. The Bethe approximation can reflect dependence of random variables (dependence between $U$ and $V$ in $\hat{p}(U, V|A)$ in our problem) to some extent. Therefore, one can intuitively expect that performance of the AMP algorithm is better than that of the variational Bayes matrix factorization, which treats $U$ and $V$ as if they were independent in $\hat{p}(U, V|A)$.

An important property of the AMP algorithm, aside from its efficiency and effectiveness, is that one can predict performance of the algorithm accurately for large-scale problems by using a set of equations, called the state evolution [6]. Analysis with the state evolution also shows that required iteration numbers are $O(1)$ even when the problem size is large. Although we can present the state evolution for the algorithm proposed in this paper and give a proof of its validity like [8, 18], we do not discuss the state evolution here due to the limited space available.

We introduce a one-parameter extension of the posterior distribution $\hat{p}(U, V|A)$ to treat the marginalization problem and the MAP problem in a unified manner. It is defined as follows:

$$\hat{p}(U, V|A; \beta) \propto \exp\left(-\frac{\beta}{2m\tau} \|A - UV^\top\|_F^2\right) \left(\hat{p}_U(U)\hat{p}_V(V)\right)^\beta, \tag{8}$$

which is proportional to $\hat{p}(U, V|A)^\beta$, where $\beta > 0$ is the parameter. When $\beta = 1$, $\hat{p}(U, V|A; \beta)$ is reduced to $\hat{p}(U, V|A)$. In the limit $\beta \to \infty$, the distribution $\hat{p}(U, V|A; \beta)$ concentrates on the maxima of $\hat{p}(U, V|A)$. An algorithm for the marginalization problem on $\hat{p}(U, V|A; \beta)$ is particularized to the algorithms for the marginalization problem and for the MAP problem for the original posterior distribution $\hat{p}(U, V|A)$ by letting $\beta = 1$ and $\beta \to \infty$, respectively. The AMP algorithm for the marginalization problem on $\hat{p}(U, V|A; \beta)$ is derived in a way similar to that described in [9], as detailed in the Supplementary Material.

In the derived algorithm, the values of variables $B_u^t = (\boldsymbol{b}_{u,1}^t, \dots, \boldsymbol{b}_{u,m}^t)^\top \in \mathbb{R}^{m \times \hat{r}}$, $B_v^t = (\boldsymbol{b}_{v,1}^t, \dots, \boldsymbol{b}_{v,N}^t)^\top \in \mathbb{R}^{N \times \hat{r}}$, $\Lambda_u^t \in \mathbb{R}^{\hat{r} \times \hat{r}}$, $\Lambda_v^t \in \mathbb{R}^{\hat{r} \times \hat{r}}$, $U^t = (\boldsymbol{u}_1^t, \dots, \boldsymbol{u}_m^t)^\top \in \mathbb{R}^{m \times \hat{r}}$, $V^t = (\boldsymbol{v}_1^t, \dots, \boldsymbol{v}_N^t)^\top \in \mathbb{R}^{N \times \hat{r}}$, $S_1^t, \dots, S_m^t \in \mathbb{R}^{\hat{r} \times \hat{r}}$, and $T_1^t, \dots, T_N^t \in \mathbb{R}^{\hat{r} \times \hat{r}}$ are calculated iteratively, where the superscript $t \in \mathbb{N} \cup \{0\}$ represents iteration numbers. Variables with a negative iteration number are defined as 0. The algorithm is as follows:

**Algorithm 2.**

$$B_{\mathrm{u}}^t = \frac{1}{m\tau}AV^t - \frac{1}{m\tau}U^{t-1}\sum_{j=1}^N T_j^t, \quad \Lambda_{\mathrm{u}}^t = \frac{1}{m\tau}(V^t)^\top V^t + \frac{1}{\beta m\tau}\sum_{j=1}^N T_j^t - \frac{1}{m\tau}\sum_{j=1}^N T_j^t, \quad (9\mathrm{a})$$

$$\boldsymbol{u}_i^t = \boldsymbol{f}(\boldsymbol{b}_{\mathrm{u},i}^t, \Lambda_{\mathrm{u}}^t; \hat{p}_{\mathbf{u}}), \qquad S_i^t = G(\boldsymbol{b}_{\mathrm{u},i}^t, \Lambda_{\mathrm{u}}^t; \hat{p}_{\mathbf{u}}), \qquad\qquad\qquad (9\mathrm{b})$$

$$B_{\mathrm{v}}^t = \frac{1}{m\tau}A^\top U^t - \frac{1}{m\tau}V^t\sum_{i=1}^m S_i^t, \quad \Lambda_{\mathrm{v}}^t = \frac{1}{m\tau}(U^t)^\top U^t + \frac{1}{\beta m\tau}\sum_{i=1}^m S_i^t - \frac{1}{m\tau}\sum_{i=1}^m S_i^t, \quad (9\mathrm{c})$$

$$\boldsymbol{v}_j^{t+1} = \boldsymbol{f}(\boldsymbol{b}_{\mathrm{v},j}^t, \Lambda_{\mathrm{v}}^t; \hat{p}_{\mathbf{v}}), \qquad T_j^{t+1} = G(\boldsymbol{b}_{\mathrm{v},j}^t, \Lambda_{\mathrm{v}}^t; \hat{p}_{\mathbf{v}}). \qquad\qquad (9\mathrm{d})$$

Algorithm 2 is almost symmetric in $U$ and $V$. Equations (9a)–(9b) and (9c)–(9d) update quantities related to the estimates of $U_0$ and $V_0$, respectively. The algorithm requires an initial value $V^0$ and begins with $T_j^0 = O$. The functions $\boldsymbol{f}(\cdot, \cdot; \hat{p}) : \mathbb{R}^{\hat{r}} \times \mathbb{R}^{\hat{r}\times\hat{r}} \to \mathbb{R}^{\hat{r}}$ and $G(\cdot, \cdot; \hat{p}) : \mathbb{R}^{\hat{r}} \times \mathbb{R}^{\hat{r}\times\hat{r}} \to \mathbb{R}^{\hat{r}\times\hat{r}}$, which have a p.d.f. $\hat{p} : \mathbb{R}^{\hat{r}} \to \mathbb{R}$ as a parameter, are defined by

$$\boldsymbol{f}(\boldsymbol{b}, \Lambda; \hat{p}) := \int \boldsymbol{u}\hat{q}(\boldsymbol{u}; \boldsymbol{b}, \Lambda, \hat{p})\mathrm{d}\boldsymbol{u}, \qquad G(\boldsymbol{b}, \Lambda; \hat{p}) := \frac{\partial \boldsymbol{f}(\boldsymbol{b}, \Lambda; \hat{p})}{\partial \boldsymbol{b}}, \qquad (10)$$

where $\hat{q}(\boldsymbol{u}; \boldsymbol{b}, \Lambda, \hat{p})$ is the normalized p.d.f. of $\boldsymbol{u}$ defined by

$$\hat{q}(\boldsymbol{u}; \boldsymbol{b}, \Lambda, \hat{p}) \propto \exp\Big(-\beta\Big(\frac{1}{2}\boldsymbol{u}^\top\Lambda\boldsymbol{u} - \boldsymbol{b}^\top\boldsymbol{u} - \log\hat{p}(\boldsymbol{u})\Big)\Big). \qquad (11)$$

One can see that $\boldsymbol{f}(\boldsymbol{b}, \Lambda; \hat{p})$ is the mean of the distribution $\hat{q}(\boldsymbol{u}; \boldsymbol{b}, \Lambda, \hat{p})$ and that $G(\boldsymbol{b}, \Lambda; \hat{p})$ is its covariance matrix scaled by $\beta$. The function $\boldsymbol{f}(\boldsymbol{b}, \Lambda; \hat{p})$ need not be differentiable everywhere; Algorithm 2 works if $\boldsymbol{f}(\boldsymbol{b}, \Lambda; \hat{p})$ is differentiable at $\boldsymbol{b}$ for which one needs to calculate $G(\boldsymbol{b}, \Lambda; \hat{p})$ in running the algorithm.

We assume in the rest of this section the convergence of Algorithm 2, although the convergence is not guaranteed in general. Let $B_{\mathrm{u}}^\infty, B_{\mathrm{v}}^\infty, \Lambda_{\mathrm{u}}^\infty, \Lambda_{\mathrm{v}}^\infty, S_i^\infty, T_j^\infty, U^\infty$, and $V^\infty$ be the converged values of the respective variables. First, consider running Algorithm 2 with $\beta = 1$. The marginal posterior distribution is then approximated as

$$\hat{p}_{i,j}(\boldsymbol{u}_i, \boldsymbol{v}_j|A) \approx \hat{q}(\boldsymbol{u}_i; \boldsymbol{b}_{\mathrm{u},i}^\infty, \Lambda_{\mathrm{u}}^\infty, \hat{p}_{\mathbf{u}})\hat{q}(\boldsymbol{v}_j; \boldsymbol{b}_{\mathrm{v},j}^\infty, \Lambda_{\mathrm{v}}^\infty, \hat{p}_{\mathbf{v}}). \qquad (12)$$

Since $\boldsymbol{u}_i^\infty$ and $\boldsymbol{v}_j^\infty$ are the means of $\hat{q}(\boldsymbol{u}; \boldsymbol{b}_{\mathrm{u},i}^\infty, \Lambda_{\mathrm{u}}^\infty, \hat{p}_{\mathbf{u}})$ and $\hat{q}(\boldsymbol{v}; \boldsymbol{b}_{\mathrm{v},j}^\infty, \Lambda_{\mathrm{v}}^\infty, \hat{p}_{\mathbf{v}})$, respectively, the posterior mean $E[UV^\top|A] = \int UV^\top \hat{p}(U, V|A)\mathrm{d}U\mathrm{d}V$ is approximated as

$$E[UV^\top|A] \approx U^\infty(V^\infty)^\top. \qquad (13)$$

The marginal MAP estimates $\boldsymbol{u}_i^{\mathrm{MMAP}}$ and $\boldsymbol{v}_j^{\mathrm{MMAP}}$ are approximated as

$$\boldsymbol{u}_i^{\mathrm{MMAP}} \approx \arg\max_{\boldsymbol{u}} \hat{q}(\boldsymbol{u}; \boldsymbol{b}_{\mathrm{u},i}^\infty, \Lambda_{\mathrm{u}}^\infty, \hat{p}_{\mathbf{u}}), \quad \boldsymbol{v}_j^{\mathrm{MMAP}} \approx \arg\max_{\boldsymbol{v}} \hat{q}(\boldsymbol{v}; \boldsymbol{b}_{\mathrm{v},j}^\infty, \Lambda_{\mathrm{v}}^\infty, \hat{p}_{\mathbf{v}}). \qquad (14)$$

Taking the limit $\beta \to \infty$ in Algorithm 2 yields an algorithm for the MAP problem (4). In this case, the functions $\boldsymbol{f}$ and $G$ are replaced with

$$\boldsymbol{f}_\infty(\boldsymbol{b}, \Lambda; \hat{p}) := \arg\min_{\boldsymbol{u}}\Big[\frac{1}{2}\boldsymbol{u}^\top\Lambda\boldsymbol{u} - \boldsymbol{b}^\top\boldsymbol{u} - \log\hat{p}(\boldsymbol{u})\Big], \quad G_\infty(\boldsymbol{b}, \Lambda; \hat{p}) := \frac{\partial \boldsymbol{f}_\infty(\boldsymbol{b}, \Lambda; \hat{p})}{\partial \boldsymbol{b}}. \qquad (15)$$

One may calculate $G_\infty(\boldsymbol{b}, \Lambda; \hat{p})$ from the Hessian of $\log\hat{p}(\boldsymbol{u})$ at $\boldsymbol{u} = \boldsymbol{f}_\infty(\boldsymbol{b}, \Lambda; \hat{p})$, denoted by $H$, via the identity $G_\infty(\boldsymbol{b}, \Lambda; \hat{p}) = (\Lambda - H)^{-1}$. This identity follows from the implicit function theorem under some additional assumptions and helps in the case where the explicit form of $\boldsymbol{f}_\infty(\boldsymbol{b}, \Lambda; \hat{p})$ is not available. The MAP estimate is approximated by $(U^\infty, V^\infty)$.

## 4.2 Properties of the algorithm

Algorithm 2 has several plausible properties. First, it has a low computational cost. The computational cost per iteration is $\mathrm{O}(mN)$, which is linear in the number of components of the matrix $A$. Calculation of $\boldsymbol{f}(\cdot, \cdot; \hat{p})$ and $G(\cdot, \cdot; \hat{p})$ is performed $\mathrm{O}(N + m)$ times per iteration. The constant

factor depends on $\hat{p}$ and $\beta$. Calculation of $\boldsymbol{f}$ for $\beta < \infty$ generally involves an $\hat{r}$-dimensional numerical integration, although they are not needed in cases where an analytic expression of the integral is available and cases where the variables take only discrete values. Calculation of $\boldsymbol{f}_\infty$ involves minimization over an $\hat{r}$-dimensional vector. When $-\log\hat{p}$ is a convex function and $\Lambda$ is positive semidefinite, this minimization problem is convex and can be solved at relatively low cost.

Second, Algorithm 2 has a form similar to that of an algorithm based on the variational Bayesian matrix factorization. In fact, if the last terms on the right-hand sides of the four equations in (9a) and (9c) are removed, the resulting algorithm is the same as an algorithm based on the variational Bayesian matrix factorization proposed in [4] and, in particular, the same as the ICM algorithm when $\beta \to \infty$. (Note, however, that [4] only treats the case where the priors $\hat{p}_{\mathbf{u}}$ and $\hat{p}_{\mathbf{v}}$ are multivariate Gaussian distributions.) Note that additional computational cost for these extra terms is $O(m + N)$, which is insignificant compared with the cost of the whole algorithm, which is $O(mN)$.

Third, when one deals with the MAP problem, the value of $C^{\mathrm{MAP}}(U, V)$ may increase in iterations of Algorithm 2. The following proposition, however, guarantees optimality of the output of Algorithm 2 in a certain sense, if it has converged.

**Proposition 1.** *Let $(U^\infty, V^\infty, S_1^\infty, \ldots, S_m^\infty, T_1^\infty, \ldots, T_N^\infty)$ be a fixed point of the AMP algorithm for the MAP problem and suppose that $\sum_{i=1}^m S_i^\infty$ and $\sum_{j=1}^N T_j^\infty$ are positive semidefinite. Then $U^\infty$ is a global minimum of $C^{\mathrm{MAP}}(U, V^\infty)$ and $V^\infty$ is a global minimum of $C^{\mathrm{MAP}}(U^\infty, V)$.*

The proof is in the Supplementary Material. The key to the proof is the following reformulation:

$$U^t = \arg\min_U \left[ C^{\mathrm{MAP}}(U, V^t) - \mathrm{tr}\left( (U - U^{t-1})\left(\frac{1}{2m\tau}\sum_{j=1}^N T_j^t\right)(U - U^{t-1})^\top \right) \right] \quad (16)$$

If $\sum_{j=1}^N T_j^t$ is positive semidefinite, the second term of the minimand is the negative squared pseudometric between $U$ and $U^{t-1}$, which is interpreted as a penalty on nearness to the temporal estimate. Positive semidefiniteness of $\sum_{i=1}^m S_i^t$ and $\sum_{j=1}^N T_j^t$ holds in almost all cases. In fact, we only have to assume $\lim_{\beta\to\infty} G(\boldsymbol{b}, \Lambda; \hat{p}) = G_\infty(\boldsymbol{b}, \Lambda; \hat{p})$, since $G(\boldsymbol{b}, \Lambda; \hat{p})$ is a scaled covariance matrix of $\hat{q}(\boldsymbol{u}; \boldsymbol{b}, \Lambda, \hat{p})$, which is positive semidefinite. It follows from Proposition 1 that any fixed point of the AMP algorithm is also a fixed point of the ICM algorithm. It has two implications: (i) Execution of the ICM algorithm initialized with the converged values of the AMP algorithm does not improve $C^{\mathrm{MAP}}(U^t, V^t)$. (ii) The AMP algorithm has not more fixed points than the ICM algorithm. The second implication may help the AMP algorithm avoid getting stuck in bad local minima.

### 4.3 Clustering via AMP algorithm

One can use the AMP algorithm for the MAP problem to perform the K-means clustering by letting $\hat{p}_{\mathbf{u}}(\boldsymbol{u}) = 1$ and $\hat{p}_{\mathbf{v}}(\boldsymbol{v}) = \hat{r}^{-1}\sum_{l=1}^{\hat{r}} \delta(\boldsymbol{v} - \boldsymbol{e}_l)$. Noting that $\boldsymbol{f}_\infty(\boldsymbol{b}, \Lambda; \hat{p}_{\mathbf{v}})$ is piecewise constant with respect to $\boldsymbol{b}$ and hence $G_\infty(\boldsymbol{b}, \Lambda; \hat{p}_{\mathbf{v}})$ is $O$ almost everywhere, we obtain the following algorithm:

**Algorithm 3** (AMP algorithm for the K-means clustering)**.**

$$B_{\mathbf{u}}^t = \frac{1}{m\tau}AV^t, \quad \Lambda_{\mathbf{u}}^t = \frac{1}{m\tau}(V^t)^\top V^t, \quad U^t = B_{\mathbf{u}}^t(\Lambda_{\mathbf{u}}^t)^{-1}, \quad S^t = (\Lambda_{\mathbf{u}}^t)^{-1}, \quad (17\mathrm{a})$$

$$B_{\mathbf{v}}^t = \frac{1}{m\tau}A^\top U^t - \frac{1}{\tau}V^t S^t, \quad \Lambda_{\mathbf{v}}^t = \frac{1}{m\tau}(U^t)^\top U^t - \frac{1}{\tau}S^t, \quad (17\mathrm{b})$$

$$\boldsymbol{v}_j^{t+1} = \arg\min_{\boldsymbol{v}\in\{\boldsymbol{e}_1,\ldots,\boldsymbol{e}_{\hat{r}}\}}\left[\frac{1}{2}\boldsymbol{v}^\top \Lambda_{\mathbf{v}}^t \boldsymbol{v} - \boldsymbol{v}^\top \boldsymbol{b}_{\mathbf{v},j}^t\right]. \quad (17\mathrm{c})$$

It is initialized with an assignment $V^0 \in \{\boldsymbol{e}_1, \ldots, \boldsymbol{e}_{\hat{r}}\}^N$. Algorithm 3 is rewritten as follows:

$$n_l^t = \sum_{j=1}^N I(\boldsymbol{v}_j^t = \boldsymbol{e}_l), \quad \tilde{\boldsymbol{u}}_l^t = \frac{1}{n_l^t}\sum_{j=1}^N \boldsymbol{a}_j I(\boldsymbol{v}_j^t = \boldsymbol{e}_l), \quad (18\mathrm{a})$$

$$l_j^{t+1} = \arg\min_{l\in\{1,\ldots,\hat{r}\}}\left[\frac{1}{m\tau}\|\boldsymbol{a}_j - \tilde{\boldsymbol{u}}_l^t\|_2^2 + \frac{2m}{n_l^t}I(\boldsymbol{v}_j^t = \boldsymbol{e}_l) - \frac{m}{n_l^t}\right], \quad \boldsymbol{v}_j^{t+1} = \boldsymbol{e}_{l_j^{t+1}}. \quad (18\mathrm{b})$$

The parameter $\tau$ appearing in the algorithm does not exist in the K-means clustering problem. In fact, $\tau$ appears because $m^{-2} \sum_{i=1}^{m} A_{ij}^2 S_i^t$ was estimated by $\tau m^{-1} \sum_{i=1}^{m} S_i^t$ in deriving Algorithm 2, which can be justified for large-sized problems. In practice, we propose using $m^{-2}N^{-1}\|A - U^t(V^t)^\top\|_F^2$ as a temporary estimate of $\tau$ at $t$th iteration. While the AMP algorithm for the K-means clustering updates the value of $U$ in the same way as Lloyd's K-means algorithm, it performs assignments of data to clusters in a different way. In the AMP algorithm, in addition to distances from data to centers of clusters, the assignment at present is taken into consideration in two ways: (i) A datum is less likely to be assigned to the cluster that it is assigned to at present. (ii) Data are more likely to be assigned to a cluster whose size at present is smaller. The former can intuitively be understood by observing that if $\boldsymbol{v}_j^t = \boldsymbol{e}_l$, one should take account of the fact that the cluster center $\tilde{\boldsymbol{u}}_l^t$ is biased toward $\boldsymbol{a}_j$. The term $2m(n_l^t)^{-1}I(\boldsymbol{v}_j^t = \boldsymbol{e}_l)$ in (18b) corrects this bias, which, as it should be, is inversely proportional to the cluster size.

The AMP algorithm for maximum accuracy clustering is obtained by letting $\beta = 1$ and $\hat{p}_{\mathrm{v}}(\boldsymbol{v})$ be a discrete distribution on $\{\boldsymbol{e}_1, \ldots, \boldsymbol{e}_{\hat{r}}\}$. After the algorithm converges, $\arg\max_{\boldsymbol{v}} \hat{q}(\boldsymbol{v}; \boldsymbol{v}_j^\infty, \Lambda_{\mathrm{v}}^\infty, \hat{p}_{\mathrm{v}})$ gives the final cluster assignment of the $j$th datum and $U^\infty$ gives the estimate of the cluster centers.

# 5  Numerical experiments

We conducted numerical experiments on both artificial and real data sets to evaluate performance of the proposed algorithms for clustering. In the experiment on artificial data sets, we set $m = 800$ and $N = 1600$ and let $\hat{r} = r$. Cluster centers $\tilde{\boldsymbol{u}}_{0,l}$, $l = 1, \ldots, r$, were generated according to the multivariate Gaussian distribution $N(\boldsymbol{0}, I)$. Cluster assignments $\boldsymbol{v}_{0,j}$, $j = 1, \ldots, N$, were generated according to the uniform distribution on $\{\boldsymbol{e}_1, \ldots, \boldsymbol{e}_r\}$. For fixed $\tau = 0.1$ and $r$, we generated 500 problem instances and solved them with five algorithms: Lloyd's K-means algorithm (K-means), the AMP algorithm for the K-means clustering (AMP-KM), the variational Bayes matrix factorization [4] for maximum accuracy clustering (VBMF-MA), the AMP algorithm for maximum accuracy clustering (AMP-MA), and the K-means++ [16]. The K-means++ updates the variables in the same way as Lloyd's K-means algorithm with an initial value chosen in a sophisticated manner. For the other algorithms, initial values $\boldsymbol{v}_j^0$, $j = 1, \ldots, N$, were randomly generated from the same distribution as $\boldsymbol{v}_{0,j}$. We used the true prior distributions of $U$ and $V$ for maximum accuracy clustering.

We ran Lloyd's K-means algorithm and the K-means++ until no change was observed. We ran the AMP algorithm for the K-means clustering until either $V^t = V^{t-1}$ or $V^t = V^{t-2}$ is satisfied. This is because we observed oscillations of assignments of a small number of data. For the other two algorithms, we terminated the iteration when $\|U^t - U^{t-1}\|_F^2 < 10^{-15}\|U^{t-1}\|_F^2$ and $\|V^t - V^{t-1}\|_F^2 < 10^{-15}\|V^{t-1}\|_F^2$ were met or the number of iterations exceeded 3000. We then evaluated the following performance measures for the obtained solution $(U^*, V^*)$:

- Normalized K-means loss $\|A - U^*(V^*)^\top\|_F^2 / (\sum_{j=1}^{N} \|\boldsymbol{a}_j - \bar{\boldsymbol{a}}\|_2^2)$, where $\bar{\boldsymbol{a}} := \frac{1}{N} \sum_{j=1}^{N} \boldsymbol{a}_j$.

- Accuracy $\max_P N^{-1} \sum_{j=1}^{N} I(P\boldsymbol{v}_j^* = \boldsymbol{v}_{0,j})$, where the maximization is taken over all $r$-by-$r$ permutation matrices. We used the Hungarian algorithm [19] to solve this maximization problem efficiently.

- Number of iterations needed to converge.

We calculated the averages and the standard deviations of these performance measures over 500 instances. We conducted the above experiments for various values of $r$.

Figure 1 shows the results. The AMP algorithm for the K-means clustering achieves the smallest K-means loss among the five algorithms, while the Lloyd's K-means algorithm and K-means++ show large K-means losses for $r \geq 5$. We emphasize that all the three algorithms are aimed to minimize the same K-means loss and the differences lie in the algorithms for minimization. The AMP algorithm for maximum accuracy clustering achieves the highest accuracy among the five algorithms. It also shows fast convergence. In particular, the convergence speed of the AMP algorithm for maximum accuracy clustering is comparable to that of the AMP algorithm for the K-means clustering when the two algorithms show similar accuracy ($r < 9$). This is in contrast to the common observation that the variational Bayes method often shows slower convergence than the ICM algorithm.

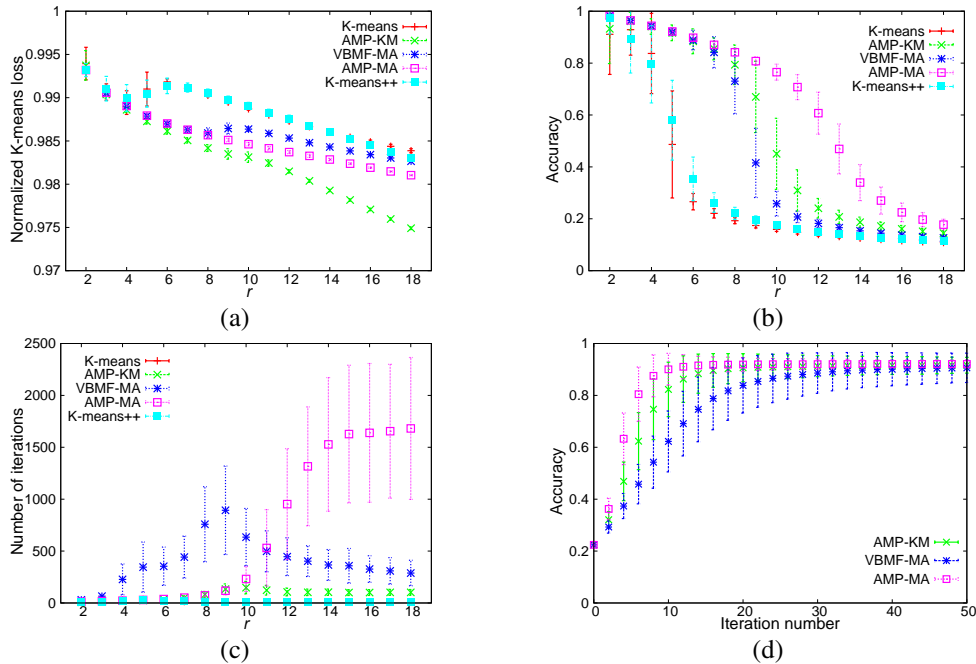

Figure 1: (a)–(c) Performance for different $r$: (a) Normalized K-means loss. (b) Accuracy. (c) Number of iterations needed to converge. (d) Dynamics for $r = 5$. Average accuracy at each iteration is shown. Error bars represent standard deviations.

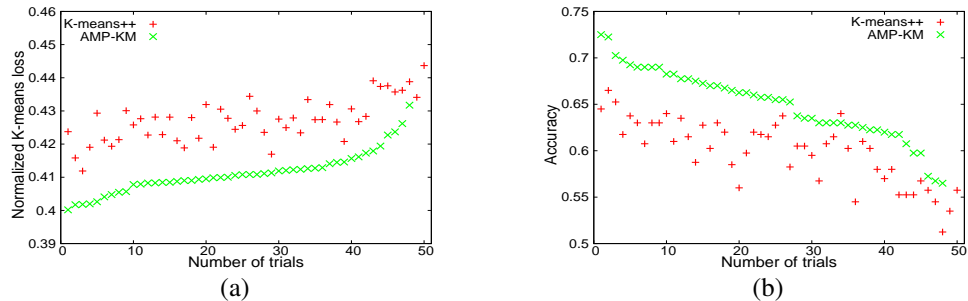

Figure 2: Performance measures in real-data experiments. (a) Normalized K-means loss. (b) Accuracy. The results for the 50 trials are shown in the descending order of performance for AMP-KM. The worst two results for AMP-KM are out of the range.

In the experiment on real data, we used the ORL Database of Faces [20], which contains 400 images of human faces, ten different images of each of 40 distinct subjects. Each image consists of $112 \times 92 = 10304$ pixels whose value ranges from 0 to 255. We divided $N = 400$ images into $\hat{r} = 40$ clusters with the K-means++ and the AMP algorithm for the K-means clustering. We adopted the initialization method of the K-means++ also for the AMP algorithm, because random initialization often yielded empty clusters and almost all data were assigned to only one cluster. The parameter $\tau$ was estimated in the way proposed in Subsection 4.3. We ran 50 trials with different initial values, and Figure 2 summarizes the results.

The AMP algorithm for the K-means clustering outperformed the standard K-means++ algorithm in 48 out of the 50 trials in terms of the K-means loss and in 47 trials in terms of the accuracy. The AMP algorithm yielded just one cluster with all data assigned to it in two trials. The attained minimum value of K-means loss is 0.412 with the K-means++ and 0.400 with the AMP algorithm. The accuracies at these trials are 0.635 with the K-means++ and 0.690 with the AMP algorithm. The average number of iterations was 6.6 with the K-means++ and 8.8 with the AMP algorithm. These results demonstrate efficiency of the proposed algorithm on real data.

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
