[Supplementary Material · nips2013_supplement.pdf]

# Supplementary Material for "Low-rank matrix reconstruction and clustering via approximate message passing"

**Ryosuke Matsushita**
NTT DATA Mathematical Systems Inc.
1F Shinanomachi Rengakan, 35,
Shinanomachi, Shinjuku-ku, Tokyo,
160-0016, Japan
matsur8@gmail.com

**Toshiyuki Tanaka**
Department of Systems Science,
Graduate School of Informatics, Kyoto University
Yoshida Hon-machi, Sakyo-ku, Kyoto-shi,
606-8501 Japan
tt@i.kyoto-u.ac.jp

## 1 The derivation of the AMP algorithm for low-rank matrix reconstruction

### 1.1 Message passing algorithm and Gaussian approximation

We derive the AMP algorithm for low-rank matrix reconstruction. More concretely, we derive the AMP algorithm that approximates $\hat{p}_{i,j}(\boldsymbol{u}_i, \boldsymbol{v}_j | A; \beta)$. We do not strive for mathematical rigor in the derivation.

We use the belief propagation message passing algorithm on the factor graph shown in Fig. 1 to approximate $\hat{p}_{i,j}(\boldsymbol{u}_i, \boldsymbol{v}_j | A; \beta)$. Every $\hat{r}$-dimensional vector $\boldsymbol{u}_i$ and $\boldsymbol{v}_j$ is represented by a variable node. Each factor node in Fig. 1 corresponds to $p(A_{i,j} | \boldsymbol{u}_i, \boldsymbol{v}_j)^\beta$. The factor nodes corresponding to $\hat{p}(\boldsymbol{u}_i)^\beta$ and $\hat{p}(\boldsymbol{v}_j)^\beta$ are omitted. At each iteration in the belief propagation message passing algorithm, the four types of probability density functions $\{\hat{\mu}^t_{(i,j)\to i}(\boldsymbol{u}_i), \mu^t_{i\to(i,j)}(\boldsymbol{u}_i), \hat{\nu}^t_{(i,j)\to j}(\boldsymbol{v}_j), \nu^{t+1}_{j\to(i,j)}(\boldsymbol{v}_j)\}$, called messages, are updated as follows:

$$\hat{\mu}^t_{(i,j)\to i}(\boldsymbol{u}_i) \propto \int \exp\Big(-\frac{\beta(A_{i,j} - \boldsymbol{u}_i^\top \boldsymbol{v}_j)^2}{2m\tau}\Big)\nu^t_{j\to(i,j)}(\boldsymbol{v}_j)\mathrm{d}\boldsymbol{v}_j, \tag{1a}$$

Figure 1: Factor graph for low-rank matrix reconstruction.

$$\mu_{i\rightarrow(i,j)}^t(\boldsymbol{u}_i) \propto \hat{p}_{\mathbf{u}}(\boldsymbol{u}_i)^\beta \prod_{l\neq j} \hat{\mu}_{(i,l)\rightarrow i}^t(\boldsymbol{u}_i), \tag{1b}$$

$$\hat{\nu}_{(i,j)\rightarrow j}^t(\boldsymbol{v}_j) \propto \int \exp\Big(-\frac{\beta(A_{i,j}-\boldsymbol{u}_i^\top\boldsymbol{v}_j)^2}{2m\tau}\Big)\mu_{i\rightarrow(i,j)}^t(\boldsymbol{u}_i)\mathrm{d}\boldsymbol{u}_i, \tag{1c}$$

$$\nu_{j\rightarrow(i,j)}^{t+1}(\boldsymbol{v}_j) \propto \hat{p}_{\mathbf{v}}(\boldsymbol{v}_j)^\beta \prod_{k\neq i} \hat{\nu}_{(k,j)\rightarrow j}^t(\boldsymbol{v}_j). \tag{1d}$$

Since these messages are defined as functions of real numbers, it is hard to implement the algorithm in the above form. A technique used in [1, 2] is to give an approximate representation of these messages in terms of some real-valued parameters. This approximation has been proved exact in the large system limit $m \rightarrow \infty$ under certain conditions. We describe this approximation in this subsection, and a further reduction in the number of parameters in the next subsection, yielding Algorithm 2 of the main paper.

We assume that $N$ goes to infinity in the same order as $m$, when we consider the limit $m \rightarrow \infty$. The order-in-probability notation $\mathrm{O}_p(\cdot)$ is used to represent approximation errors for large $m$.

**Definition 1.** Let $\{X_m\}$ be sequences of random variables taking values in $\mathbb{R}^d$ and $\{a_m\}$ be a sequence of positive reals. Then $X_m = \mathrm{O}_p(a_m)$ if and only if for any $\epsilon > 0$, there exists a constant $C(\epsilon)$ and an integer $M(\epsilon)$ such that if $m \geq M(\epsilon)$, then

$$\Pr\Big[\frac{\|X_m\|}{a_m} > C(\epsilon)\Big] \leq \epsilon. \tag{2}$$

We write $X_{i,j,m} = \mathrm{O}_p(a_m)$ for random variables $X_{i,j,m}, i = 1,\ldots,m, j = 1,\ldots,N_m$, if $X_{i_m,j_m,m} = \mathrm{O}_p(a_m)$ for an arbitrary sequence $\{(i_m,j_m)\}$ that satisfies $1 \leq i_m \leq m$ and $1 \leq j_m \leq N_m$. We assume that $\mu_{i\rightarrow(i,j)}^t$ and $\nu_{j\rightarrow(i,j)}^t$ have finite moments of sufficiently high orders. We also assume that these moments are $\mathrm{O}_p(1)$ and follow distributions with finite moments of sufficiently high orders. (Here, randomness is with respect to $A$). We further assume that every row of $U_0$ and $V_0$ follows a distribution with finite moments of sufficiently high orders. The equations $A_{i,j} = W_{i,j} + \boldsymbol{u}_{0,i}^\top\boldsymbol{v}_{0,j} = \mathrm{O}_p(m^{1/2})$ and $A_{i,j}^2 = \mathrm{O}_p(m)$ are used in the following.

By substituting (1a) into the product on the right-hand side of (1b), one obtains

$$\prod_{l\neq j} \hat{\mu}_{(i,l)\rightarrow i}^t(\boldsymbol{u}_i) \propto \int \exp\Big(-\frac{\beta\sum_{l\neq j}(A_{i,l}-\boldsymbol{u}_i^\top\boldsymbol{v}_l)^2}{2m\tau}\Big) \prod_{l\neq j} \nu_{l\rightarrow(i,l)}^t(\boldsymbol{v}_l)\mathrm{d}\boldsymbol{v}_l. \tag{3}$$

The right-hand side is the expectation of $\exp\big(-(2m\tau)^{-1}\beta\sum_{l\neq j}(A_{i,l}-\boldsymbol{u}_i^\top\boldsymbol{v}_l)^2\big)$ with respect to the independent random variables $\boldsymbol{v}_l \sim \nu_{l\rightarrow(i,l)}^t(\boldsymbol{v}_l)$, $l = 1, \ldots, N$. Since $z := \sum_{l\neq j}\frac{\beta}{2m\tau}(A_{i,l}-\boldsymbol{u}_i^\top\boldsymbol{v}_l)^2$ is a sum of $(N-1)$ independent random variables $\frac{\beta}{2m\tau}(A_{i,l}-\boldsymbol{u}_i^\top\boldsymbol{v}_l)^2$, $l \neq j$, for given $A$, it can be approximated by a Gaussian random variable for large $N$ due to the central limit theorem. The mean and variance of $z$ are

$$\bar{z} = \frac{\beta}{2m\tau}\sum_{l\neq j}\big(A_{i,l}^2 - 2A_{i,l}\boldsymbol{u}_i^\top\,\mathbb{E}[\boldsymbol{v}_l] + \boldsymbol{u}_i^\top\,\mathbb{E}[\boldsymbol{v}_l\boldsymbol{v}_l^\top]\boldsymbol{u}_i\big), \tag{4}$$

and

$$\frac{\beta^2}{m^2\tau^2}\sum_{l\neq j}\Big(A_{i,l}^2\boldsymbol{u}_i^\top\big(\mathbb{E}[\boldsymbol{v}_l\boldsymbol{v}_l^\top] - \mathbb{E}[\boldsymbol{v}_l]\,\mathbb{E}[\boldsymbol{v}_l^\top]\big)\boldsymbol{u}_i$$

$$- A_{i,l}(\mathbb{E}[(\boldsymbol{u}_i^\top\boldsymbol{v}_l)^3] - \mathbb{E}[\boldsymbol{u}_i^\top\boldsymbol{v}_l]\,\mathbb{E}[(\boldsymbol{u}_i^\top\boldsymbol{v}_l)^2]) + \frac{1}{4}(\mathbb{E}[(\boldsymbol{u}_i^\top\boldsymbol{v}_l)^4] - \mathbb{E}[(\boldsymbol{u}_i^\top\boldsymbol{v}_l)^2]^2)\Big), \tag{5}$$

respectively. Each summand in

$$\frac{\beta^2}{m^2\tau^2}\sum_{l\neq j}\Big(-A_{i,l}(\mathbb{E}[(\boldsymbol{u}_i^\top\boldsymbol{v}_l)^3] - \mathbb{E}[\boldsymbol{u}_i^\top\boldsymbol{v}_l]\,\mathbb{E}[(\boldsymbol{u}_i^\top\boldsymbol{v}_l)^2]) + \frac{1}{4}(\mathbb{E}[(\boldsymbol{u}_i^\top\boldsymbol{v}_l)^4] - \mathbb{E}[(\boldsymbol{u}_i^\top\boldsymbol{v}_l)^2]^2)\Big) \tag{6}$$

is $\mathrm{O}_p(m^{1/2})$ and therefore (6) is $\mathrm{O}_p(m^{-1/2})$. (In fact, one can conclude that (6) is $\mathrm{O}_p(m^{-1})$ if one assumes that the dependence among the summands is weak.) We define $\sigma^2$ on the basis of this

observation by

$$\sigma^2 := \frac{\beta^2}{m^2\tau^2} \sum_{l \neq j} A_{i,l}^2 \boldsymbol{u}_i^\top \big(\mathbb{E}[\boldsymbol{v}_l \boldsymbol{v}_l^\top] - \mathbb{E}[\boldsymbol{v}_l]\,\mathbb{E}[\boldsymbol{v}_l^\top]\big)\boldsymbol{u}_i, \tag{7}$$

and suppose $z \sim N(\bar{z}, \sigma^2)$. Replacing the integral on the right-hand side of (3) by the expectation with respect to $z$, one obtains

$$\prod_{l \neq j} \hat{\mu}_{(i,l)\to i}^t(\boldsymbol{u}_i) \propto \frac{1}{\sqrt{2\pi\sigma^2}} \int \exp(-z) \exp\Big(-\frac{(z-\bar{z})^2}{2\sigma^2}\Big)\mathrm{d}z \tag{8}$$

$$= \exp\Big(-\bar{z} + \frac{\sigma^2}{2}\Big), \tag{9}$$

where the last equation is derived by performing the Gaussian integral in (8). Let the mean and covariance matrix of $\boldsymbol{v}_j \sim \nu_{j\to(i,j)}^t(\boldsymbol{v}_j)$ be $\boldsymbol{v}_{j\to(i,j)}^t$ and $\beta^{-1}T_{j\to(i,j)}^t$, respectively. Then (9) is rewritten as

$$\prod_{l \neq j} \hat{\mu}_{(i,l)\to i}^t(\boldsymbol{u}_i) \propto \exp\big(-\frac{\beta}{2}\boldsymbol{u}_i^\top \Lambda_{\mathrm{u},i\to(i,j)}^t \boldsymbol{u}_i + \beta \boldsymbol{u}_i^\top \boldsymbol{b}_{\mathrm{u},i\to(i,j)}^t\big), \tag{10}$$

where $\boldsymbol{b}_{\mathrm{u},i\to(i,j)}^t$ and $\Lambda_{\mathrm{u},i\to(i,j)}^t$ are defined by

$$\boldsymbol{b}_{\mathrm{u},i\to(i,j)}^t := \frac{1}{m\tau} \sum_{l \neq j} A_{i,l} \boldsymbol{v}_{l\to(i,l)}^t, \tag{11}$$

and

$$\Lambda_{\mathrm{u},i\to(i,j)}^t := \frac{1}{m\tau} \sum_{l \neq j} \Big(\boldsymbol{v}_{l\to(i,l)}^t (\boldsymbol{v}_{l\to(i,l)}^t)^\top + \frac{1}{\beta}T_{l\to(i,l)}^t - \frac{A_{i,l}^2}{m\tau}T_{l\to(i,l)}^t\Big), \tag{12}$$

respectively. It follows from (10) and (1b) that

$$\mu_{i\to(i,j)}^t(\boldsymbol{u}_i) = \hat{q}(\boldsymbol{u}_i; \boldsymbol{b}_{\mathrm{u},i\to(i,j)}^t, \Lambda_{\mathrm{u},i\to(i,j)}^t, \hat{p}_{\mathrm{u}}), \tag{13}$$

where $\hat{q}$ is defined in (11) of the main paper. From (11)–(13), one can see that $\mu_{i\to(i,j)}^t$ depends on the p.d.f. $\nu_{l\to(i,l)}^t$ only through its mean $\boldsymbol{v}_{l\to(i,l)}^t$ and covariance $\beta^{-1}T_{l\to(i,l)}^t$. Via a similar argument, one can represent the p.d.f. $\nu_{j\to(i,j)}^{t+1}$ with the mean $\boldsymbol{u}_{i\to(i,j)}^t$ and the covariance $\beta^{-1}S_{i\to(i,j)}^t$ of $\mu_{i\to(i,j)}^t$:

$$\boldsymbol{b}_{\mathrm{v},j\to(i,j)}^t := \frac{1}{m\tau} \sum_{k \neq i} A_{k,j} \boldsymbol{u}_{k\to(k,j)}^t, \tag{14}$$

$$\Lambda_{\mathrm{v},j\to(i,j)}^t := \frac{1}{m\tau} \sum_{k \neq i} \Big(\boldsymbol{u}_{k\to(k,j)}^t (\boldsymbol{u}_{k\to(k,j)}^t)^\top + \frac{1}{\beta}S_{k\to(k,j)}^t - \frac{A_{k,j}^2}{m\tau}S_{k\to(k,j)}^t\Big), \tag{15}$$

$$\nu_{j\to(i,j)}^{t+1}(\boldsymbol{v}_j) = \hat{q}(\boldsymbol{v}_j; \boldsymbol{b}_{\mathrm{v},j\to(i,j)}^t, \Lambda_{\mathrm{v},j\to(i,j)}^t, \hat{p}_{\mathrm{v}}). \tag{16}$$

The variables $\boldsymbol{u}_{i\to(i,j)}^t$, $S_{i\to(i,j)}^t$, $\boldsymbol{v}_{j\to(i,j)}^{t+1}$, and $T_{j\to(i,j)}^{t+1}$ are calculated by

$$\boldsymbol{u}_{i\to(i,j)}^t = \boldsymbol{f}\big(\boldsymbol{b}_{\mathrm{u},i\to(i,j)}^t, \Lambda_{\mathrm{u},i\to(i,j)}^t; \hat{p}_{\mathrm{u}}\big), \tag{17a}$$

$$S_{i\to(i,j)}^t = G\big(\boldsymbol{b}_{\mathrm{u},i\to(i,j)}^t, \Lambda_{\mathrm{u},i\to(i,j)}^t; \hat{p}_{\mathrm{u}}\big), \tag{17b}$$

$$\boldsymbol{v}_{j\to(i,j)}^{t+1} = \boldsymbol{f}\big(\boldsymbol{b}_{\mathrm{v},j\to(i,j)}^t, \Lambda_{\mathrm{v},j\to(i,j)}^t; \hat{p}_{\mathrm{v}}\big), \tag{17c}$$

$$T_{j\to(i,j)}^{t+1} = G\big(\boldsymbol{b}_{\mathrm{v},j\to(i,j)}^t, \Lambda_{\mathrm{v},j\to(i,j)}^t; \hat{p}_{\mathrm{v}}\big), \tag{17d}$$

where $\boldsymbol{f}$ and $G$ are defined in (10) of the main paper. We recall that $\boldsymbol{f}(\boldsymbol{b}, \Lambda; \hat{p})$ is the mean of the p.d.f. $\hat{q}(\boldsymbol{u}; \boldsymbol{b}, \Lambda, \hat{p})$ and that $G(\boldsymbol{b}, \Lambda; \hat{p})$ is its covariance matrix scaled by $\beta$. Thus (1a)–(1d), which are equations for updating p.d.f.s, are reduced to ones for updating their means and covariances.

Our algorithm, as said above, updates only some real-valued parameters and this is common to the expectation propagation algorithm [3]. The ideas behind these algorithms, however, are different. The expectation propagation algorithm assumes that messages are in the exponential family and that they have specified parametric forms. In the derivation of our algorithm, we do not assume any parametric form of the messages and have deduced that the mean and covariance are sufficient to update the messages by assuming the large system limit $m \to \infty$ and using the central limit theorem.

## 1.2 Reducing the number of the messages

The message passing algorithm (17a)–(17d) involves $4mN$ messages. In this section, using another technique introduced in [1, 2], we reduce the number of the messages to $2(m + N)$. From (14) and (15), one can see that the dependence of $\boldsymbol{b}^t_{\mathrm{v},j\to(i,j)}$ and $\Lambda^t_{\mathrm{v},j\to(i,j)}$ on $i$ is weak. In fact, the variables defined by

$$\boldsymbol{b}^t_{\mathrm{v},j} := \frac{1}{m\tau} \sum_{k=1}^{m} A_{k,j} \boldsymbol{u}^t_{k\to(k,j)}, \tag{18}$$

$$\Lambda^t_{\mathrm{v},j} := \frac{1}{m\tau} \sum_{k=1}^{m} \left( \boldsymbol{u}^t_{k\to(k,j)} (\boldsymbol{u}^t_{k\to(k,j)})^\top + \frac{1}{\beta} S^t_{k\to(k,j)} - \frac{A^2_{k,j}}{m\tau} S^t_{k\to(k,j)} \right), \tag{19}$$

$$\boldsymbol{v}^{t+1}_j := \boldsymbol{f}\big(\boldsymbol{b}^t_{\mathrm{v},j}, \Lambda^t_{\mathrm{v},j}; \hat{p}_{\mathbf{v}}\big), \quad T^{t+1}_j := G\big(\boldsymbol{b}^t_{\mathrm{v},j}, \Lambda^t_{\mathrm{v},j}; \hat{p}_{\mathbf{v}}\big), \tag{20}$$

which are independent of $i$, satisfy the following equations:

$$\boldsymbol{b}^t_{\mathrm{v},j\to(i,j)} = \boldsymbol{b}^t_{\mathrm{v},j} - \frac{1}{m\tau} A_{i,j} \boldsymbol{u}^t_{i\to(i,j)}$$
$$= \boldsymbol{b}^t_{\mathrm{v},j} + \mathrm{O}_p(m^{-1/2}), \tag{21}$$

$$\Lambda^t_{\mathrm{v},j\to(i,j)} = \Lambda^t_{\mathrm{v},j} - \frac{1}{m\tau} \left( \boldsymbol{u}^t_{i\to(i,j)} (\boldsymbol{u}^t_{i\to(i,j)})^\top + \frac{1}{\beta} S^t_{i\to(i,j)} - \frac{A^2_{i,j}}{m\tau} S^t_{i\to(i,j)} \right)$$
$$= \Lambda^t_{\mathrm{v},j} + \mathrm{O}_p(m^{-1}), \tag{22}$$

$$\boldsymbol{v}^{t+1}_{j\to(i,j)} = \boldsymbol{v}^{t+1}_j + \mathrm{O}_p(m^{-1/2}), \tag{23}$$

$$T^{t+1}_{j\to(i,j)} = T^{t+1}_j + \mathrm{O}_p(m^{-1/2}), \tag{24}$$

where the last two equations hold under an appropriate assumption of smoothness of $\boldsymbol{f}(\boldsymbol{b}, \Lambda; \hat{p}_{\mathrm{v}})$ and $G(\boldsymbol{b}, \Lambda; \hat{p}_{\mathrm{v}})$. We define $\boldsymbol{b}^t_{\mathrm{u},i}, \Lambda^t_{\mathrm{u},i}, \boldsymbol{u}^t_i$ and $S^t_i$ via equations analogous to (18)–(20). As shown by (21)–(24), the variables defined here, said to be singly-indexed, approximate the doubly-indexed variables such as $\boldsymbol{b}^t_{\mathrm{v},j\to(i,j)}$ with negligible errors for large $m$. The number of the singly-indexed variables is $\mathrm{O}(m)$ and it is less than that of the doubly-indexed ones, which is $\mathrm{O}(m^2)$. The goal in the following is to derive equations that updates the singly-indexed variables without using doubly-indexed ones.

Equations (20) involves only variables independent of $i$ and have the same form as (9d) of the main paper. In the following, we represent the right-hand sides of (18) and (19) in terms of $\boldsymbol{u}^t_k$, $S^t_k$, and $\boldsymbol{v}^t_j$, which will turn out to be necessary, instead of $\boldsymbol{u}^t_{k\to(k,j)}$ and $S^t_{k\to(k,j)}$. This representation should be exact in non-vanishing order. Our desired representation of the right-hand side of (19) is obtained simply by replacing $\boldsymbol{u}^t_{k\to(k,j)}$ and $S^t_{k\to(k,j)}$ with $\boldsymbol{u}^t_k$ and $S^t_k$, respectively:

$$\Lambda^t_{\mathrm{v},j} = \frac{1}{m\tau} \sum_{k=1}^{m} \left( \boldsymbol{u}^t_k (\boldsymbol{u}^t_k)^\top + \frac{1}{\beta} S^t_k - \frac{A^2_{k,j}}{m\tau} S^t_k \right) + \mathrm{O}_p(m^{-1/2}), \tag{25}$$

where we used the equations $\boldsymbol{u}^t_{k\to(k,j)} = \boldsymbol{u}^t_k + \mathrm{O}_p(m^{-1/2})$ and $S^t_{k\to(k,j)} = S^t_k + \mathrm{O}_p(m^{-1/2})$. This simple replacement does not work for (18) because the terms of the order $\mathrm{O}_p(m^{-1/2})$ in $\boldsymbol{u}^t_{k\to(k,j)}$ can contribute to the summation in (18) in non-vanishing order. One, therefore, has to evaluate $\boldsymbol{u}^t_{k\to(k,j)}$ exactly in up to $\mathrm{O}_p(m^{-1/2})$. For this evaluation, we use the Taylor expansion of $\boldsymbol{f}(\boldsymbol{b}, \Lambda; \hat{p}_{\mathbf{u}})$ with respect to $(\boldsymbol{b}^t_{\mathrm{u},i\to(i,j)}, \Lambda^t_{\mathrm{u},i\to(i,j)})$ around $(\boldsymbol{b}^t_{\mathrm{u},i}, \Lambda^t_{\mathrm{u},i})$. Noting that $\boldsymbol{b}^t_{\mathrm{u},i\to(i,j)} = \boldsymbol{b}^t_{\mathrm{u},i} - (m\tau)^{-1} A_{i,j} \boldsymbol{v}^t_{j\to(i,j)}$ and $\Lambda^t_{\mathrm{u},i\to(i,j)} = \Lambda_{\mathrm{u},i} + \mathrm{O}_p(m^{-1})$, one obtains

$$\boldsymbol{u}^t_{i\to(i,j)} = \boldsymbol{f}\big(\boldsymbol{b}^t_{\mathrm{u},i}, \Lambda^t_{\mathrm{u},i}; \hat{p}_{\mathbf{u}}\big) - \frac{A_{i,j}}{m\tau} \frac{\partial \boldsymbol{f}}{\partial \boldsymbol{b}^t_{\mathrm{u},i}}\big(\boldsymbol{b}^t_{\mathrm{u},i}, \Lambda^t_{\mathrm{u},i}; \hat{p}_{\mathbf{u}}\big) \boldsymbol{v}^t_{j\to(i,j)} + \mathrm{O}_p(m^{-1})$$

$$= \boldsymbol{u}^t_i - \frac{A_{i,j}}{m\tau} S^t_i \boldsymbol{v}^t_j + \mathrm{O}_p(m^{-1}). \tag{26}$$

Substituting (26) into (18) gives

$$\boldsymbol{b}^t_{\mathrm{v},j} = \frac{1}{m\tau} \sum_{k=1}^{m} A_{k,j} \boldsymbol{u}^t_k - \frac{1}{m^2\tau^2} \sum_{k=1}^{m} A^2_{k,j} S^t_k \boldsymbol{v}^t_j + \mathrm{O}_p(m^{-1/2}), \tag{27}$$

which is our desired representation.

For further simplification, we replace $A_{k,j}^2$ appearing in (25) and (27) with its expectation

$$\mathbb{E}[(\boldsymbol{u}_k^\top \boldsymbol{v}_j + W_{k,j})^2] = \mathbb{E}[W_{k,j}^2] + \mathbb{E}[(\boldsymbol{u}_k^\top \boldsymbol{v}_j)^2]$$
$$= m\tau + \mathrm{O}(1). \tag{28}$$

Since the variance of $(m\tau)^{-1} A_{k,j}^2 S_k^t$ is $\mathrm{O}(1)$, the variance of the arithmetic mean $m^{-1} \sum_{k=1}^m (m\tau)^{-1} A_{k,j}^2 S_k^t$ is $\mathrm{O}(m^{-1})$ if we assume that the summands are weakly dependent with each other. This observation justifies replacing $A_{i,l}^2$ with its expectation. This replacement results in

$$\boldsymbol{b}_{\mathrm{v},j}^t = \frac{1}{m\tau} \sum_{k=1}^m A_{k,j} \boldsymbol{u}_k^t - \frac{1}{m\tau} \sum_{k=1}^m S_k^t \boldsymbol{v}_j^t + \mathrm{O}_p(m^{-1/2}), \tag{29}$$

and

$$\Lambda_{\mathrm{v},j}^t = \Lambda_{\mathrm{v}}^t + \mathrm{O}_p(m^{-1/2}), \tag{30}$$

$$\Lambda_{\mathrm{v}}^t := \frac{1}{m\tau} \sum_{k=1}^m \Big( \boldsymbol{u}_k^t (\boldsymbol{u}_k^t)^\top + \frac{1}{\beta} S_k^t - S_k^t \Big). \tag{31}$$

These are used to update $\boldsymbol{b}_{\mathrm{v},j}^t$ and $\Lambda_{\mathrm{v}}^t$ if terms of vanishing order in $m \to \infty$ are ignored, giving (9c) of the main paper. One can calculate $\boldsymbol{b}_{\mathrm{u},i}^t$ and $\Lambda_{\mathrm{u}}^t$ in a similar way, yielding (9a) of the main paper.

Finally, we calculate the marginal distribution $\hat{p}_{i,j}(\boldsymbol{u}_i, \boldsymbol{v}_j | A; \beta)$. When the iteration has converged, the marginal distribution is calculated by

$$\hat{p}_{i,j}(\boldsymbol{u}_i, \boldsymbol{v}_j | A; \beta) \propto \exp\Big( -\frac{\beta(A_{i,j} - \boldsymbol{u}_i^\top \boldsymbol{v}_j)^2}{2m\tau} \Big) \mu_{i \to (i,j)}^\infty(\boldsymbol{u}_i) \nu_{j \to (i,j)}^\infty(\boldsymbol{v}_j)$$
$$= \exp\Big( -\frac{\beta(A_{i,j} - \boldsymbol{u}_i^\top \boldsymbol{v}_j)^2}{2m\tau} \Big) \hat{q}_\beta(\boldsymbol{u}_i; \boldsymbol{b}_{\mathrm{u},i}^\infty, \Lambda_{\mathrm{u}}^\infty, \hat{p}_{\mathbf{u}}) \hat{q}_\beta(\boldsymbol{v}_j; \boldsymbol{b}_{\mathrm{v},j}^\infty, \Lambda_{\mathrm{v}}^\infty, \hat{p}_{\mathbf{v}}). \tag{32}$$

Because the first factor can be ignored in the limit $m \to \infty$, one obtains

$$\hat{p}_{i,j}(\boldsymbol{u}_i, \boldsymbol{v}_j | A; \beta) \propto \hat{q}_\beta(\boldsymbol{u}_i; \boldsymbol{b}_{\mathrm{u},i}^\infty, \Lambda_{\mathrm{u}}^\infty, \hat{p}_{\mathbf{u}}) \hat{q}_\beta(\boldsymbol{v}_j; \boldsymbol{b}_{\mathrm{v},j}^\infty, \Lambda_{\mathrm{v}}^\infty, \hat{p}_{\mathbf{v}}). \tag{33}$$

## 2 The proof of Proposition 1

We give the proof of Proposition 1 of the main paper.

*Proof.* The update of $U^t$ in the AMP algorithm for the MAP problem is rewritten as

$$U^t = \arg\min_U \Big[ \sum_{i=1}^m \Big( \frac{1}{2} \boldsymbol{u}_i^\top \Lambda_{\mathrm{u}}^t \boldsymbol{u}_i - \boldsymbol{u}_i^\top \boldsymbol{b}_{\mathrm{u},i}^t - \log \hat{p}_{\mathrm{u}}(\boldsymbol{u}_i) \Big) \Big]$$

$$= \arg\min_U \Big[ \mathrm{tr}\Big( \frac{1}{2} U \Lambda_{\mathrm{u}}^t U^\top - B_{\mathrm{u}}^t U^\top \Big) - \sum_{i=1}^m \log \hat{p}_{\mathrm{u}}(\boldsymbol{u}_i) \Big]$$

$$= \arg\min_U \Big[ \frac{1}{2m\tau} \mathrm{tr}\Big( U(V^t)^\top V^t U^\top - U \sum_{j=1}^N T_j^t U^\top \Big)$$
$$\quad - \frac{1}{m\tau} \mathrm{tr}\Big( A V^t U^\top - U^{t-1} \sum_{j=1}^N T_j^t U^\top \Big) - \sum_{i=1}^m \log \hat{p}_{\mathrm{u}}(\boldsymbol{u}_i) \Big]$$

$$= \arg\min_U \Big[ \frac{1}{2m\tau} \| A - U(V^t)^\top \|_F^2 - \mathrm{tr}\Big( (U - U^{t-1})(\frac{1}{2m\tau} \sum_{j=1}^N T_j^t)(U - U^{t-1})^\top \Big)$$
$$\quad - \sum_{i=1}^m \log \hat{p}_{\mathrm{u}}(\boldsymbol{u}_i) \Big]$$

$$= \arg\min_{U} \left[ C^{\mathrm{MAP}}(U, V^t) - \mathrm{tr}\left( (U - U^{t-1})(\frac{1}{2m\tau} \sum_{j=1}^{N} T_j^t)(U - U^{t-1})^\top \right) \right], \qquad (34)$$

where the third equation follows from (9a) of the main paper and the last equation follows from (5) of the main paper. Therefore, the following inequality holds for any $U \in \mathbb{R}^{m \times \hat{r}}$:

$$C^{\mathrm{MAP}}(U^\infty, V^\infty) \leq C^{\mathrm{MAP}}(U, V^\infty) - \mathrm{tr}\left( (U - U^\infty)(\frac{1}{2m\tau} \sum_{j=1}^{N} T_j^t)(U - U^\infty)^\top \right). \qquad (35)$$

Since $\sum_{j=1}^{N} T_j^\infty$ is positive semidefinite, the right-hand side is bounded from above by $C^{\mathrm{MAP}}(U, V^\infty)$, proving the first half of the proposition. The second half can be proved similarly. □