[Reviews · NeurIPS 2013]

Submitted by Assigned_Reviewer_2

The authors present an algorithm for matrix reconstruction under noisy observations. The particular setting looks at low-rank matrices with additional assumptions on the factors and uses an Approximate Message Passing approach in order to speed up the classical, computationally expensive, Bayesian approach. The authors also connect matrix reconstruction with K-means clustering, which is an interesting application domain for the proposed algorithms.

To the best of my knowledge, the Approximate Message Passing approach for matrix reconstruction is novel and interesting. The connections between low-rank matrix factorizations and K-means are fairly well-known (e.g., PCA provides a factor two approximation algorithm for K-means). However, this allows the authors to provide a nice experimental evaluation of their algorithms and compare them to k-means and k-means ++. Interestingly, their approach seems faster and more efficient than classical k-means and k-means ++ according to their empirical data. The authors compare both the Frobenius norm residual, as well as the actual clusterings, which is a nice feature of their experimental evaluations.

The main weak point of the paper is that the proposed algorithm comes with few theoretical guarantees in terms of convergence. This is to be expected, since many other algorithms for K-means also have only weak properties. The authors might want to at least cite more papers in Theoretical Computer Science that provide provably accurate algorithms for the K-means objective function.
Summary: A solid paper presenting Approximate Message Passing algorithms for low-rank matrix reconstruction and k-means. Promising experimental evaluation, somewhat weak theoretical results.

Submitted by Assigned_Reviewer_4

Review of "Low-rank matrix reconstruction and clustering"

This paper contributes a new algorithm for low-rank matrix reconstruction which is based on an application of Belief Propagation (BP) message-passing to a Bayesian model of the reconstruction problem. The algorithm, as described in the "Supplementary Material", incorporates two simplifying approximations, based on assuming a large number of rows and columns, respectively, in the input matrix. The algorithm is evaluated in a novel manner against Lloyd's K-means algorithm by formulating clustering as a matrix reconstruction problem. It is also compared against Variational Bayes Matrix Factorization (VBMF), which seems to be the only previous message-passing reconstruction algorithm.

Cons

There are some arguments against accepting the paper. Because a new algorithm is being evaluated on a non-standard problem (clustering encoded as matrix factorization), it is not easy to interpolate the experimental results to predict how the algorithm would perform on more conventional matrix reconstruction problems. For instance, two references appear to be cited for VBMF, which are Lin and Teh (2007); and Raiko, Ilin and Karhunen (2007). Both of these papers use the Netflix dataset to evaluate their algorithm against predecessors. It would be ideal if the present paper had used the same dataset. Although BP is usually more accurate than Variational, evaluating the present BP variant using a new criterion creates doubt surrounding its actual competitiveness. Even if Netflix or a similar dataset can't be used, the authors should explain in the paper why this is the case.

The algorithm itself appears to be a more or less straightforward application of BP to a problem which had been previously addressed with Variational Message Passing. Although new, it is not exactly groundbreaking. The most interesting part of it, to me, is the approximations which are introduced in the limit $N\to\infty$ and $m\to\infty$, where $m \times N$ is the dimensions of the input matrix. However, the validity of these approximations, which are only described in supplementary material, is never directly tested, and I think they could be explained a bit more clearly.

There are some serious problems regarding the citation of prior work. When I first read the paper, I thought that it was introducing the application of matrix factorization to clustering as an original contribution. The text of the abstract and Section 2.2 give this impression. I felt betrayed when I learned from other reviewers that the connection is well-known. I don't see a good reason why the paper would not make this clear to the reader. If it is too well-known to cite any particular reference, then one should just say that it is well-known. Otherwise, cite prior work.

Also, EP should be cited, since that is usually the name people give to applying BP to continuous classes of distributions, and the relationship with EP to the paper's algorithm should be explained. Relatedly, the main algorithm is most plainly understood as an application of Belief Propagation, but this fact is not mentioned until Section 4.1. It should be mentioned in the abstract.

Pros

The paper was interesting to read, and presents a new and potentially useful algorithm. The mathematics of the paper was possible to follow, and although I did not replicate it by hand, I got the sense that it would be possible to do so. Although I think it is reasonable to be suspicious of new evaluation criteria, the K-means problem may be sufficiently general to give a fair comparison of the algorithms, and certainly shows a benefit for the new algorithm in the experiments.

Clarity

I found the early presentation fairly easy to follow. The introduction was clear, as was the summary of earlier work. The fact that the derivation of the algorithm only appears in the Supplemental Material is a drawback. I wish that the derivation could be outlined in the main paper. The experiments section was clear, and although some of the plots show up poorly in black and white, they were still readable.

I found it difficult to understand the description of the algorithm, and I was not able to check the correctness of the derivation. The mathematics was the weakest part of this presentation. Even at a very superficial level it was difficult to parse. For instance, I don't understand why factors of $1 / m \tau_w$ appear before each term in equations 9a, 9b, 9d, and 9e. These could be factored out, to make the expressions easier to read. Also, the last two terms in 9b and 9e, respectively, could be combined.

There seem to be an excess of hats, tildes, subscripts, and superscripts. For instance, there is a $\tau_w$ but no $\tau$, why not just replace $\tau_w$ by $\tau$? Also, the most important Section 4 contains no p, q, or r, but only \hat{p}, \hat{q}, \hat{r} - why not give readers a break and say at the beginning of the section "We'll be dropping the hats here for brevity"? And the 't' superscripts in the messages seem to be wholly unnecessary. When the left hand side has a "t+1" you just need to change "=" to "\gets" and then you can drop all of the t's. I have a hard time imagining that the algorithm was originally derived by the authors using such cumbersome notation. I would suggest going back to the first notation you used and looking to see if it is simpler.

The exposition could be improved: Why not explain that equation 5 is the negative logarithm of equation 2? Or that equation 8 is just equation 2 to the power of $\beta$? Algorithm 2 seems to be almost row-column symmetrical, why not point this out? And even make it half as long, by saying, "then copy and paste these equations, switching u and v"?

The meaning of functions in 10 should be explained near their definition. It would be clearer to say "f_\beta is the mean of q_\beta", rather than giving an equation; but if you decide to give an equation, then why not say what it means? At the end of Section 4.2, it says "G_\beta(b, \Lambda; p) is a scaled covariance matrix of q_\beta(u; b, \Lambda, p)", but this was not obvious at first, and wasn't even mentioned at the definition of G.

None of the messages have \beta subscripts, even though they depend on f and G which have \beta subscripts. But f and G don't depend directly on \beta, only on q_\beta. So it's not clear why these subscripts are propagated only as far as f and G and no further. I would suggest eliminating them entirely, even from q. The parameter \beta is simply a global variable, which is just fine. Before (15) and (16), it will just be necessary to say something like "in the limit \beta \to \infty, f and G take the following form:".

On a deeper level, there were other things about the algorithm that I would like to understand better. What is the significance of the m factor in the additive noise variance? Does it play a role in applications of matrix factorization? Does it play a role in the approximation used to derive the algorithm? It seems to be the only thing making the algorithm fail to be row-column symmetrical, is this true? I think the authors know the answer to these questions, but do not comment on them.

Other questions:

Is it standard to use a tilde to denote column vectors? I didn't understand this at first.

Just curious - why is N capitalized, but m lowercase?

In section 5, I would like a citation after VBMF-MA to indicate the primary reference guiding the implementation of this competing algorithm. There are two citations for variational matrix factorization appearing earlier in the paper, and it is not clear which is intended (or if it is both).

In the supplementary material: For equation 5, I think it would be clearer to write a product of exponentials for the two terms involving $z$, to make it more obvious that one is the pdf. For section 2, I had trouble with the step from equation 14 to 15, regarding big-O of $m$ terms. I am not sure if I just need to think harder, but this is where I got stuck.

Also in the supplementary material, at the top of page 2 it says "A technique used in ... is to give an approximate representation of these message in terms of a few number of real-valued parameters". Without reading these references, I am not sure how exactly the approximation described below this text relates to what has been published before. It would be good to clarify the novelty of the algorithm's derivation in the text itself, and even in the main paper.

The English is very good, and the meaning always gets across, but there are some places where it reads like it was written by someone not entirely skilled in the use of articles. This can be distracting for some readers. Even in the title - I think it should be something like "Low-rank matrix reconstruction and clustering using an approximate message passing algorithm". For other places, I will make a list of suggestions which may be of use to the authors:

p. 1

"Since properties" -> "Since the properties"

"according to problems" -. "according to the problem"

"has flexibility" - "has enough flexibility"

"motivate use of" -> "motivate the use of"

p. 2

"We present results" -> "We present the results"

p. 3

"We regard that" -> "We see that"

"maximum accuracy clustering" could be italicized

"Previous works" -> "Previous work"

"ICM algorithm; It" -> "ICM algorithm: it"

p. 4

"particularized" -> "specialized"

p. 5

"plausible properties" -> "discernable properties"?

p. 6

"stuck to bad local minima" -> "stuck in bad local minima"?

p. 7

"This is contrastive to that" -> "This is in contrast to the fact that"

The Supplementary Material also has some language issues, for instance:

p. 2

"of these message" -> "of these messages"

"a few number" -> "a few"

I liked the diagram in Figure 1 in the Supplementary Material, which I found very helpful. If it is possible to create more diagrams, that would add to the paper.
Summary: The mathematics seems interesting, and the algorithm should be published somewhere - as the first effort to apply belief propagation to matrix factorization, it fills a certain important role. But the paper needs more work before it can be accepted. The paper is sufficiently lacking in clarity and scholarship as to put an unacceptable burden on readers, which would reflect poorly on a conference which accepted it in its present state.

Submitted by Assigned_Reviewer_5

This paper proposes an approximate message passing algorithm for matrix factorization. After showing that the matrix factorization problem can be seen as a generalization of clustering problem, the message passing algorithm derived for matrix factorization is specialized to clustering. Lastly, experiments were conducted to compare the proposed algorithm with the k-means++ algorithm.

Quality: Mathematical derivations in this paper are very sketchy and omits many non-trivial steps even in appendix.

1) In line 139 of Appendix, why is the second term of (11) O(1/\sqrt(m))? \beta seems to be omitted in Section 2.
2) Some of the analysis, including derivation of (19), seems to assume that m and n grows in the same order. After all, asymptotic analysis in this section are a bit terse and I am concerned about its mathematical rigor.
3) In line 167, why is m \tau_w^2 the expectation of A_{il}^2? A is observed data, and m\tau_w^2 is only variance of the likelihood; the rational for such approximation should be verified.
4) How would the message passing algorithm derived for finite \beta converge to the MAP problem? Message passing algorithm minimizes the KL divergence, but \lim_\beta \min KL = \min \lim_\beta KL does not necessarily hold.
5) The proof of Proposition 1 is too brief for me. How is (17) derived?

Clarity: The paper is well-structured and it is not difficult to grasp what is the main point of the paper.

Originality: Since authors are mostly concerned about estimating first and second moments of the marginal posterior, isn't the whole algorithm just application of Expectation-Propagation (EP) to matrix factorization? EP was already used in matrix factorization in the following paper: http://research.microsoft.com/pubs/79460/www09.pdf Section 1 of Appendix seems to be standard EP updated procedures; please correct me if I am wrong.

Also, it seems that there should be an interesting connection between the clustering algorithm proposed by authors and the clutter problem of original EP paper: http://research.microsoft.com/en-us/um/people/minka/papers/ep/minka-ep-uai.pdf , if the likelihood of clutter problem is switched to the uniform mixture of gaussian distributions. Just to clarify, in this paragraph I am just suggesting a connection and not attacking the originality of the paper.

Significance: Authors mainly focus on its application to clustering, but considering the vast amount of literature on clustering, comparing only to k-means++ may not be sufficient to prove the practical usefulness of the algorithm.
Summary: Proofs in this paper, even in appendix, are very sketchy and thus it is hard to evaluate its technical correctness. Also, I suspect that it is an application of expectation-propagation to matrix factorization problem, which was already done by Stern et al. http://research.microsoft.com/pubs/79460/www09.pdf
Author Feedback

Author rebuttal: Responses to Assigned_Reviewer_4:
1) When I first read the paper, I thought that it was introducing the application of matrix factorization to clustering as an original contribution. [...] Otherwise, cite prior work.

It is true that the connection between matrix factorization and clustering is well known, as found in:
Ding et al., "Convex and Semi-Nonnegative Matrix Factorizations", IEEE Trans. PAMI, vol.32, pp.45-55, 2010,
but it should be noted that the above paper proposes soft versions of K-means clustering only, which do not explicitly consider hard constraints on class indicators.
We should thus add the sentence
"Although the idea of applying low-rank matrix factorization to clustering is not new [citation], this paper is, to our knowledge, the first one that proposes an algorithm that explicitly deals with the constraint that each datum should be assigned to just one cluster in the framework of low-rank matrix factorization."
in the last paragraph in Introduction, citing the above reference.
We should also cite this paper at the beginning of Sect. 2.2.
These changes will make this paper's contribution clearer.

2) EP should be cited, since that is usually the name people give to applying BP to continuous classes of distributions, and the relationship with EP to the paper's algorithm should be explained.

Please see 9) in Responses to Assigned_Reviewer_5.

3) the main algorithm is most plainly understood as an application of Belief Propagation, but this fact is not mentioned until Section 4.1. It should be mentioned in the abstract.

We should change the sentence in the abstract
"We propose an efficient approximate message passing algorithm to perform ..."
to
"We propose an efficient approximate message passing algorithm derived from belief propagation algorithm to perform ...".

4) Why not explain that equation 5 is the negative logarithm of equation 2? Or that equation 8 is just equation 2 to the power of $\beta$? Algorithm 2 seems to be almost row-column symmetrical, why not point this out?
5) The meaning of functions in 10 should be explained near their definition.

We should make changes in accordance with these comments, which will make the meaning of the equations clearer.

6) What is the significance of the m factor in the additive noise variance? [...] I think the authors know the answer to these questions, but do not comment on them.

We thank the reviewer's careful consideration.
In short, the m factor introduces the "proper" scaling in the limit m, N to infinity, playing a role in the derivation of the algorithm:
For example, the fact that A_{il}^2 is of order m is used in the derivation of equation 4 in Supplementary Material.
Additionally, without the m factor, the contribution of the prior distribution p_V(V) is vanishingly small compared with the likelihood p(A|U,V) when m grows.
To make these points clearer, we should add the sentence
"The factor m in the noise variance plays a role in the derivation of the algorithm, where we assume that m and N go to infinity in the same order."
to the first paragraph in Sect. 2.1.

7)
We thank the suggestions on notation and English.
In accordance with the reviewer's comments, we should make changes including
i) replacing \tau_w with \tau.
ii) removing \beta subscript from f, G, and q.
These changes will make the paper easier to read.

Responses to Assigned_Reviewer_5:
8) Mathematical derivations in this paper are very sketchy and omits many non-trivial steps even in appendix.
8-2) Some of the analysis, including derivation of (19), seems to assume that m and n grows in the same order. After all, asymptotic analysis in this section are a bit terse and I am concerned about its mathematical rigor.

We should add to the beginning of line 63 in Supplementary Material:
"We assume that m and N grow in the same order."

8-3) In line 167, why is m \tau_w^2 the expectation of A_{il}^2? A is observed data, and m\tau_w^2 is only variance of the likelihood; the rational for such approximation should be verified.

We thank the reviewer for correcting us.
We should write in line 167 in Supplementary Material:
"replacing A_{il}^2 with its expectation E[(u_iv_l + w_{il})^2]=E[w_{il}^2] + E[(u_iv_l)^2] = m\tau_w + O(1)".

9) Since authors are mostly concerned about estimating first and second moments of the marginal posterior, isn't the whole algorithm just application of Expectation-Propagation (EP) to matrix factorization? EP was already used in matrix factorization in the following paper: http://research.microsoft.com/pubs/79460/www09.pdf Section 1 of Appendix seems to be standard EP updated procedures; please correct me if I am wrong.

We thank the comment pointing out the relationship between EP and our algorithm.
It is true that both our algorithm and EP update only a finite number of moments, not a full distribution. They are, however, different.
The expectation propagation algorithm assumes that messages are in the exponential family and that they have specified parametric forms.
In the derivation of our algorithm, we do not assume any parametric form of the messages. We have deduced that the mean and covariance are sufficient to update the messages by using the central limit theorem under the assumption that m and N go to infinity.
We should add a paragraph describing the difference between EP and our algorithm at the end of Section 1 of Supplementary Material not to mislead the reader.
(Although describing it in the main paper would be better, we cannot do it since space is limited.)